# Review of the Heat Stress-Induced Responses in Dairy Cattle

**DOI:** 10.3390/ani13223451

**Published:** 2023-11-09

**Authors:** Claudia Giannone, Marco Bovo, Mattia Ceccarelli, Daniele Torreggiani, Patrizia Tassinari

**Affiliations:** Department of Agricultural and Food Sciences (DISTAL), Alma Mater Studiorum University of Bologna, Viale Fanin 48, 40127 Bologna, Italy; claudia.giannone2@unibo.it (C.G.); mattia.ceccarelli5@unibo.it (M.C.); daniele.torreggiani@unibo.it (D.T.); patrizia.tassinari@unibo.it (P.T.)

**Keywords:** heat stress, THI, dairy cow, cattle, animal response

## Abstract

**Simple Summary:**

Next-generation numerical approaches, such as machine learning techniques and big data analytics, are also increasingly applied in the animal production sector. Still today, one of the most investigated matters in the dairy cow sector is the detection and the evaluation of the effects induced by heat stress condition. This review provides, in a single document, an overview, as complete as possible, of the heat stress-induced responses in dairy cattle aiming to transfer the wide veterinary knowledge available in the literature to researchers and technicians who are developing numerical models and decision support system tools.

**Abstract:**

In the dairy cattle sector, the evaluation of the effects induced by heat stress is still one of the most impactful and investigated aspects as it is strongly connected to both sustainability of the production and animal welfare. On the other hand, more recently, the possibility of collecting a large dataset made available by the increasing technology diffusion is paving the way for the application of advanced numerical techniques based on machine learning or big data approaches. In this scenario, driven by rapid change, there could be the risk of dispersing the relevant information represented by the physiological animal component, which should maintain the central role in the development of numerical models and tools. In light of this, the present literature review aims to consolidate and synthesize existing research on the physiological consequences of heat stress in dairy cattle. The present review provides, in a single document, an overview, as complete as possible, of the heat stress-induced responses in dairy cattle with the intent of filling the existing research gap for extracting the veterinary knowledge present in the literature and make it available for future applications also in different research fields.

## 1. Introduction

The key elements leading the dairy sector in the most recent years and probably in the years to come, in a general context that requires increasing sustainability but at the same time respecting animal needs, aim to increase milk production and quality, ensure the health and welfare of cows, improve the efficiency in the use of resources and the reduction in gaseous emissions and the recovery and reuse of livestock manure. Animal welfare is strictly related to sustainability due to the consequences in terms of milk quantity and quality, which affect the efficiency of the use of natural resources. For this purpose, a crucial point is the prevention of heat stress, as it markedly jeopardizes animal welfare in several countries of the Mediterranean area. Heat stress in dairy cows can be defined as the stress induced in the cattle when they are unable to dissipate heat without modifying the body thermal balance. Heat stress is usually related to environmental conditions, for example, in animals reared in environment characterized by high temperature, high humidity or exposed to strong solar radiation [1,2,3]. In few cases, heat stress can be attributed to an internal heat overproduction by the animal [4,5]. The thermo-neutral zone of dairy cows (i.e., the air temperature range within which the heat production by the animal is at the minimum and the amount of energy available for milk production is at the maximum) is in the range of 5–25 °C [6]. Outside of this range, the animal can be unable to dissipate metabolically produced or absorbed heat and thermal balance cannot be maintained, and as a result, animal welfare can be negatively affected [7]. All this entails animal responses induced by heat stress, which can be classified in physiological responses, morphological responses, behavioral responses, metabolic responses, productive responses, an immune status responses [8]. The different classes are the object of specific description in the following sections of the paper.

Due to its spread in several regions, heat stress is probably one of the most investigated problems in the dairy sector, in a worldwide context, and it is attracting more and more the attention of researchers and farmers. In confirmation, Figure 1 shows the bibliography trend, in a 30-year timespan, investigating the heat stress in dairy cows as extracted by Scopus by filtering for title, keywords and abstract. An interesting aspect to underline is that, if the first research works had mainly an experimental approach, based on veterinary observations or laboratory tests, recently, publication of an increasing number of papers and research based on numerical simulations and statistical analyses on the problem was observed, with the main object to define, numerically, possible correlations or dependences between environmental, production, behavior and welfare data.

On the one hand, the real-time, or almost real-time, detection of the animals suffering heat stress is not a trivial matter since most of the available technologies and proposed methodologies are usually applied only in particular conditions, e.g., by using specific data not commonly available and collected in commercial farms. On the other hand, it is important to separate the problem of the detection of the cow heat stress condition (referred to a short-term response arising in the same instant of heat load occurrence) from the assessment of the medium- and long-term responses of the animal. In fact, a heat stress condition can be recognized as a deviation from normality of (particular) animal-based indicators like, for example, the respiration rate, the body temperature or the plasma cortisol concentrations. Instead, the medium- or long-term responses most commonly observed in dairy cattle exposed to prolonged heat stress are the reduction in the feed intake [9,10,11], in rumination time [9,12,13], in milk production and milk quality [14,15,16,17], in welfare and health, a detriment [18,19] of the reproductive performance [20,21] and immune response [22,23] and a modification of the daily activity parameters, e.g., resting time and number of steps [1].

As stated, a first generation of studies investigated from an experimental point of view, based on veterinary observations or laboratory tests, the problem of heat stress in dairy cows. More recently, due to both an increasing number of sensors used for daily livestock monitoring and the introduction of more modern technologies supporting the breeders, there has been a strong increase in quantity and quality of data daily collected in the farms which has made it possible to apply statistical methods and numerical modelling techniques in a more robust and extensive way. In fact, Precision Livestock Farming (PLF) approaches based on the acquisition of precise and real-time data concerning production and welfare of individual animals and monitoring systems for animals and environmental conditions are becoming common tools in daily herd management. The availability of a large dataset has provided the conditions for the application of advanced numerical techniques already applied in other fields of research.

As a matter of fact, more recently, several papers investigated the relation between environmental conditions and one or more animal-based indicators with the main object of modelling short- or long-term effects of heat stress. In most of these works, the environmental conditions were modelled by the Temperature–Humidity Index (THI) in the barn [24] or indices derived by the THI, like the Heat Load Index (HLI), developed mainly for animals raised outdoors and considering also air velocity and solar radiation values [1,25,26,27,28,29,30,31,32,33]. Other authors adopted different indicators like the Black Globe Humidity Index (BGHI) or Comprehensive Climate Index (CCI). Following the literature, probably the most widely used predictive model is THI, even if THI does not entirely represent the thermal environment. Therefore, THI values can only act as a rough indicator for the effects of heat stress, in lieu of knowing the internal body temperature of animals.

As far as the animal-based measures are concerned, a large part of the research directly focused on milk yield and tried to establish numerical models for the assessment of the milk yield reduction or for the evaluation of the time lag between heat stress condition and production drop [34]. Others instead used more invasive measures like plasma cortisol, or temperature in the rectum, vagina, or rumen [35,36,37], as alternative, less invasive proxy measures, e.g., cortisol metabolites in feces and milk, panting rhythm, or external body temperature [18,38,39,40,41].

Driven by the rapid increase in studies and papers based on PLF approaches, several of which refer to numerical machine learning (ML) techniques, we present in the following paper the results of a literature review aiming to summarize the main effects induced by heat stress on dairy cows. All this is performed with the main objective of collecting in a coordinated and comprehensive way the different animal-based measures that can be used as predictive features of the presence of heat stress in future numerical approaches and modelling studies. In fact, studies that are based on ML techniques and making use of big data approaches are rapidly increasing and are receiving more and more consent and attention from the world of research, but at the same time there could be the risk of dispersing the information potential represented by the physiological animal component. The latter should maintain its central role for a correct development of these models and tools, which are becoming more and more used by farmers and practitioners for the management of the animals, especially in the dairy sector. The remainder of this review is structured as follows. The second section provides the review methodology and describes the database of the papers analyzed in the work. The sections from the third to the eighth present, in the order the physiological, the morphological, the behavioral, the metabolic, the productive, the immune system responses of dairy cows to heat stress solicitation. Section nine is devoted to depicting the main directions for future research according to the documents reviewed.

## 2. Review Methodology

### 2.1. Criteria for the Bibliographic Source Selection

For the aim of the present work, the literature review approach, as defined in [42], was considered also in accordance with the main frame reported in [43]. In fact, the literature review approach seems to be appropriate for the scope of the review aiming to enucleate the major findings of the reviewed papers, summarize in a complete framework the measures mostly adopted for the detection and the assessment of heat stress effects, provide indications on the possibility of using these measures in future research based on data-driven models and advanced ML approaches. The bibliographic research was realized by searching in the Web of Science, ScienceDirect and Scopus databases and selecting the search field “Article title”, “Abstract”, “Keywords” options. The string combinations used were as follows: (“heat stress” OR ‘‘heat load” OR “temperature humidity index” OR “THI”) AND (‘‘dairy cow” OR “dairy cattle” OR “dairy herd” OR “Holstein”). No geographical or time limitations were applied during the searches. The published time of the literature in this work is up to January 2023, when the search ended. Once the search in the databases was completed, the duplicated documents were removed. At the end of this procedure, a first raw database of 2175 documents was identified. In the further selection stages, only peer-reviewed papers, congress abstracts, and book chapters written in English were taken into consideration and, after the careful reading of the full text, articles not pertaining to the scope of the review were discarded. Figure 2 shows the main steps of the selection process.

It is worth noticing that measures usually adopted for animal welfare assessment can be grouped into three main classes, i.e., animal-based measures (ABM), resource-based measures (RBM), and management-based measures (MBM) as summarized and commented on in Table 1. With reference to this categorization and for the scope of the present review, the most common reason for exclusion from the first dataset was the absence in the discharged paper of quantitative information related to animal-based measures. In fact, several papers used RBM and MBM for the evaluation of animal welfare, but this case is out of the scope of the present review paper.

Therefore, finally, 120 documents were selected and thoroughly reviewed and the results contribute to populate the following sections of the present review. Figure 3a displays the timeline distribution of the 120 selected papers starting from 1962. The number of publications shows an increasing trend with a peak in 2020. Moreover, Figure 3b illustrates the geographical distribution of the selected papers considering the location of the studied animals.

It is worth noting that investigations related to heat stress assessment and evaluation are not equally distributed worldwide, but on the contrary, most of the studies originate in few countries, which, in the order the ten countries with the highest numbers of papers, are USA, Italy, India, Germany, Brazil, China, Australia, Spain, Israel and Canada. This approximately reflects the worldwide distribution of milk production crossed with the presence of several days in the year characterized by high temperatures and/or humidity values.

### 2.2. Review of the Reviews: Progress, Overlaps and Research Trends

The final database, collecting 120 papers selected for the purpose of the present work, includes 19 review type articles. They are briefly summarized below for the sake of completeness. In 2002 [45], a review article was published discussing the concept of thermoneutral zone, heat production and heat gain, heat dissipation mechanisms, and how high-producing lactating dairy cow responds to heat stress. The review provided a list of factors influencing heat stress in lactating cows and reports the main physical, metabolic, and production responses. The paper reported a comparison between trends of productive data in the USA and Israel from about 1940 to 1990 and concludes that adequate supply of nutrients including a balanced mixture of dietary minerals and genetic progress would play the pivotal role in the modern dairies. In a review paper published in 2003 [46], the two pathways by which heat stress leads to infertility of dairy cattle were deeply described and commented on.

The authors provided three viable solutions for alleviate heat-related infertility: by intensively using the cooling systems, by the provision of high-quality forage and feed to overcome negative energy balance and using hormonal treatments to induce normal cyclicity. In the work published by Atrian and Shahryar [47], the authors reviewed the metabolic disorders correlated to heat stress and provided a general picture of the most efficient managemental methods and practices, as well as nutritional ways for the prevention the heat stress.

Moreover, in paper [48], the authors indicated that heat-stressed animals employ a new systemic physiology to direct metabolic and fuel selection priorities independent of nutrient intake or energy balance. Review paper [49] collected and synthesized the main information about the impact of heat stress on health, production and reproduction in dairy animals. The collated aspects are related to feed intake and rumen physiology, acid-base balance, impacts on the immune system, production and reproduction performance of heat-stressed dairy animals. Paper [50] had the objective to assess the decline in performances of reproductive traits such as service period, conception rate and pregnancy rate of dairy cattle and buffaloes with respect to increase in temperature humidity index in temperate, tropical or subtropical climates. In [51], the authors reviewed the effects of heat stress on specific fertility problems of the ovarian function, focusing on the impairment of the follicle-enclosed oocyte since in the female reproductive tract the ovarian pool of oocytes is highly sensitive to hyperthermia.

The objective of review [10] was to take a broad approach to assessing the effects of heat stress on dairy cow welfare by using the three pillars related to (1) the biological functioning (and health) of the animal, (2) the affective states and (3) the naturalness of its life under the current management practices. The authors observed that evidence suggests that changes in milk composition may be more useful than milk yield decrease to assess cows in immediate heat stress. They concluded that gaps in the literature highlight the need for research into the pain, frustration, aggression, and malaise associated with heat stress. Ref. [52] reviewed the ways in which heat stress impacts both milk yield and composition. The review provided an insight into the ways in which heat stress affects production and elucidates the mechanisms through which the reduced production is brought about while an animal, exposed to heat stress challenges, also produces milk with reduced protein, fat, and solids-not-fat. Ref. [53] reviewed the available cooling applications and assessed the potential use in other regions to reduce the stressful heat exposure and limit the negative effects on health and performance of dairy cows, and the considerable economic losses for the sector. In a parallel way, in [1], a review was presented of the state of knowledge on the topic of the most widely used environmental methods used for determining and predicting heat stress in dairy cows analyzing the most popular climate indexes. Then, in [54], the authors collated and synthesized information pertaining to livestock adaptation to heat stress and advanced technologies available to quantify heat stress responses in livestock. The most investigated biological makers of both phenotypic and genotypic origin were identified by the analysis of the literature. In review paper [55], the authors compared different methods and equipment available for measuring body temperature and its fluctuations since these parameters are key indicators of health and well-being in animals, and moreover, these parameters have diagnostic potential especially regarding heat stress-related diseases.

Following the outcomes in [56], respiratory rate represents one of the most sensitive physiological animal-based indicators for detecting and monitoring heat stress. In the review, the authors discussed the thermal conservative and heat dissipating roles of the respiration rate with the main aim of enhancing its adoption in evaluating both cold and heat animal adaptation. Then, paper [57] reviewed the main studies about the physiological and productive changes due to heat stress in cattle with the object to suggest and develop effective measures to mitigate the impact of heat load on animals. On the other hand, review [6] summarized information collected on dairy cattle heat stress over the years, with reference to production, reproduction, nutrition, health, and welfare with updates till 2020. In [58], the authors summarized the progress achieved in the development of indices that may be readily applied. Thermal comfort indices were compared, as well as their subsequent application in field studies highlighting difference between conditions in the laboratory versus those encountered in field studies.

The authors concluded that monitoring systems, big data analyses and artificial intelligence algorithms are needed in the future for real-time assessment and minimization of heat stress. Then, paper [59] reviewed the stages of pathogenic progression that can bring to lethal heat stress in dairy cattle. They observed that death frequently occurs days following the extreme heat event, when temperatures are seemingly no longer threatening, and furthermore, death due to heat stress is almost never recognized as a differential diagnosis but is often misled. Review paper [60] summarized three different approaches used as predictive models: bioclimatic indexes, machine learning, and mechanistic models. The work also focused on the application of the current knowledge as algorithms to be used for the management of climatic control systems.

The careful review of the 19 papers, as expected, highlighted the existence of some overlaps between the papers, not only in terms of citations but also in terms of reviewed arguments and suggestions paving the bases for the perspective research. The temporal comparison of the review papers, covering the years from 2002 to 2022, showed the different milestones of the research in the field and highlighted a shift of the objectives of the research from the problems of defining and quantifying the effects and the diseases caused by the heat stress to the themes of the calibration of numerical modelling, identification of early warnings and prediction of veterinary diseases. It is worth to note that Figure 4 shows the temporal network map of the keywords most frequently used by the 19 review papers. The dimension of the node increases with the number of occurrences while the different color indicates the years with most of the occurrences. As the figure shows, the most recent trends of the research are moving towards the introduction of numerical approaches and models that are unavoidably linked to the need to have robust and reliable datasets available.

### 2.3. Why This Further Review Paper on Heat Stress?

As said above, the recent trend of studies in the dairy cow heat stress research field requires a huge effort related to the definition of the most informative veterinary parameters and aspects to use for forecasting models and decision-support systems. The need to have, in a single document, an overview of the heat stress-induced responses in dairy cattle has encouraged the writing of this review paper.

A comprehensive view is today missing in the literature, but the emerging numerical technologies need to be fed with robust and informative dataset. The main aim is to fill the existing research gap for extracting the veterinary knowledge already present in the literature on cow responses to heat stress and make the results available to those who are developing numerical models and approaches.

The present article analyzes and includes works that cover aspects of the animal response. Since the phenomenon of heat stress is strongly linked to the microclimate in the barn and to the susceptibility of the individual animal, the indications provided in this review must be integrated with key elements that, for brevity reasons, cannot be reported here. Therefore, the present work must be seen in a broader context and, as often happens when approaching interdisciplinary problems, it needs to be related to the specific literature of the other pieces of the puzzle. The information organized in the present review is preparatory to the definition of the most suitable numerical tools.

## 3. Physiological Responses

### 3.1. Body Temperature and Respiration Rate

Heat stress is a prominent issue among dairy cattle, primarily due to their heightened vulnerability to fluctuations in surrounding temperature, which is attributed to their increased metabolic rate that is required to produce milk [60]. When dairy cattle experience heat stress, body temperature and respiration rate can be affected [61], resulting in health consequences [8]. In fact, heat stress significantly damages the thermal homeostasis system of the cows and alters body temperature. In study [62], the research findings shed light on the significant impact of heat stress on the thermal homeostasis system of heifers. Heat stress, in this context, refers to a state in which animals experience difficulties in maintaining their normal body temperature as a result of exposure to intense heat. When the surrounding air temperature surpasses 30 °C, it exerts a considerable strain on the heifers’ ability to regulate their body temperature. This manifests in notable increases in both ocular (eye) and skin surface temperatures, which are indicative of their struggle to dissipate excess heat. Similarly, ref. [63] investigated the applicability of THI for playing a crucial role in evaluating and quantifying heat stress in crossbred dairy animals, particularly in regions with temperate to hot climates. The THI is a composite index that takes into account both temperature and humidity levels, offering a comprehensive measure of the environmental heat load. When the THI value exceeds 80, it signifies a particularly high level of heat stress. Meanwhile, crossbreed cattle only experienced slight to no stress when the THI was below 74, whereas a THI between 74 and 79 resulted in a moderate level of stress.

In agreement with the previous findings, ref. [64] observed that the increment in the internal animal temperature of Angus and Taurine breed adapted (e.g., Romosinuano) bulls exposed to heat stress was induced by a heat load greater than its dissipation capacity. This result was in line with the finding in [54]. In fact, animal production was negatively impacted significantly earlier than the attainment of the critical tolerance threshold because of the body’s difficulty in maintaining a constant temperature. Ref. [18] investigated the effects of heat stress on body temperature in healthy dairy cows early postpartum in two weather periods with distinctly different THIs (i.e., 59.8 ± 3.8 and 74.1 ± 4.4). When a cow perceives thermal discomfort, it begins to alter behavior and endures physiological adaptations [65]. According to investigations in [66], an increase in core body temperature is strongly correlated to the amount of time the cows spend standing in a 24 h period.

Study [58] showed that as the heat load index increases, cows spend less time lying down; the respiration rates and body temperature increase. Indeed, aside from alterations in the temperature of the body, heat stress can lead to a rise in the respiratory rate in dairy cows [67]. As the THI rises, the cattle begin to pant trying to cool the body [68,69]. Panting is a form of the evaporative cooling process that is closely connected to heat stress [10] and, as a quick and severe reaction to heat stress, this behavior occurs in environments with high THI levels [7].

Furthermore, heavy breathing promotes increased CO_2_ exhalation, which contributes to rumen acidosis. In fact, the body should maintain a fixed HCO_3_ to CO_2_ ratio (i.e., 20:1) to operate as an efficient blood pH buffering mechanism [70]. Due to the reduction in blood CO_2_ caused by hyperventilation, the kidney secretes HCO_3_ in order to maintain this ratio. This limits the HCO_3_ quantity available for buffering and sustaining the rumen pH [70].

### 3.2. Heart Rate and Rumination Time

Monitoring dairy cattle pulse rates can be a useful method for detecting animals affected by heat stress and at risk of developing more significant health issues. With specific reference to heart rate, in [71], the authors investigated the impact of thermal stress in Holstein bull calves during a warm episode in summer to evaluate the immediate physiological reactions of calves to heat stress. Under heat load circumstances, animals housed in shaded pens have a lower heart rate than non-protected animals. This result was supported in [72] for dairy cows in and out of shade. The pulse rate of the animals in the sunlight was much greater than the pulse rate of cows within a shelter. These findings agree with those reported in [62], where it was confirmed that ambient temperature affects heart rate. When the surrounding temperature was higher than 30 °C, heifers under heat stress had a higher heart rate with a 15% rise in the rate of beats per minute. The daily evaluation of native zebu breeds also revealed greater intensities for physiological indicators including heart rate and respiratory rate throughout the afternoon [73]. Similarly, Murrah buffaloes raised in the Indian area of Punjab showed greater pulse rate, respiration rate, and rectal temperature during hot–humid climate conditions [74].

As far as the rumination time in dairy cows is concerned, it has been demonstrated that heat stress can have a significant impact. Several studies have shown that high temperatures reduce the cattle rumination time [75] which is a vital component of the digestive process [76] and plays an important role in the milk yield process [77]. As an example, the study conducted in [78] identified a negative correlation between the THI and the rumination time in Holstein dairy cows. The amount of time spent ruminating decreases as THI rises. As part of this experimental research, six distinct THI thresholds were employed for the categorize instances of thermal stress: “safe” (THI < 68), “mild discomfort” (68 ≤ THI < 72), “discomfort” (72 ≤ THI < 75), “alert” (75 ≤ THI < 79), “danger” (79 ≤ THI < 84), and “emergency“ (THI ≥ 84). This result was confirmed in [13], in which the authors carried out research on Italian Friesian dairy cows raised in a free stall barn, identifying a THI value of 52 as the threshold at which ruminating time decreases.

Furthermore, in ref. [79], the authors reported a rumination time reduction of about 2 min for every daily maximum THI point beyond the threshold value of 76 in 21 Italian Holstein cows. Likewise, in ref. [69], the authors noticed an inverse relationship between rumination time and THI. They identified a decrease in ruminating time during the night under heat stress circumstances. During colder hours, when there is a decrease in the heat stress effects, animals tend to allocate a greater amount of time to ruminating. Dairy cows under heat stress may minimize ruminating activity during cooler hours of the day to efficiently decrease the metabolic temperature [69]. A similar finding was evidenced in [12], which focused on the effect of elevated temperatures on time and daily pattern of rumination. In that study, the authors assessed that the rumination time is affected by heat stress, and they suggested that the modifications caused by reduced rumination time exerted an effect on the kinetics of ruminal feed degradation, and that heat stress resulted in a reduction in both the rate and extent of in situ feed degradability.

### 3.3. Reproduction and Fertility

Elevated temperatures can significantly impact reproduction and fertility of dairy cattle [80]. Cows undergo physiological changes when exposed to severe temperatures, which can result in decreased fertility and reproduction potential. When the body temperature increases, the endocrine system suffers, causing hormones to become unbalanced. The impact of heat stress on dairy cows has detrimental effects on their reproductive cycle [81], such as causing premature corpus luteum death, resulting in an augment of infertility and abortion. Furthermore, high temperatures can increase the risk of fetal death. In fact, in ref. [82], the authors observed that progesterone levels and prolactin concentrations were adversely associated with the estrus cycle. In addition to the impacts of extreme heat stress, lower plasma progesterone levels were also a result of hyperprolactinemia, which may have impeded follicular development before ovulation and limited luteal tissue growth after ovulation, resulting in an infertility. Moreover, in ref. [83], the authors deduced that the prolonged luteal phases, evident in heat stress-affected cattle, resulted from insufficient estradiol secretion by a follicle likely impaired by heat stress.

With specific regard to the ovarian pool of oocytes, it is one of the components of the female reproductive tract most vulnerable to heat [51]. High temperatures pose a significant threat to the ovarian pool of oocytes, affecting gene regulation in oocytes and early embryos [84]. According to research in [51], because of their protracted growth process, ovarian follicles and their encapsulated eggs are susceptible to external heat stress throughout the hot months. As a result, heat-induced changes in the early phases of follicular development might manifest later as impaired oocyte maturation and developmental competence. This mechanism might explain why cow fertility decreases in the fall when the cows are no longer subjected to heat stress, implying a carryover impact on the oocyte.

Furthermore, thermal damage to oocytes causes morphological abnormalities, oxidative stress, nuclear fragmentation and mitochondrial dysfunction [85]. In ref. [86], the authors noted the proportion of oocytes reaching the Blastocystis stage was lower in hot–warm seasons compared to that in cool seasons. Heat stress inhibits the production of the luteinizing hormone, estradiol, and the pre-ovulatory surge, which results in poor follicle development and inactive ovaries in cattle [87]. Although it is susceptible to maternal heat stress, the pre-implantation embryo becomes less sensitive as it develops. Heat stress was observed to reduce the fraction of embryos that progressed to the blastocyst stage in lactating cows on Day 1 following estrus, when embryos are one to two cells [88]. Indeed, even mild maternal hyperthermia caused by ambient thermal stress can impair reproductive activities such as estrus, fertilization, embryonic growth and survival [89].

With respect to the conception rate (CR), it has been noted that lactating dairy cows experience a decline throughout the summer season [90]. In study [30], the CR was highly related to the daily average THI on the day of breeding. CR fell steadily, beginning at the lowest THI threshold of 41 and increasing by one unit for each point increase in the average THI on the day of breeding. These data suggest that heat stress occurs in lactating dairy cows also at low THI values and in moderate climates, as indicated by a reduction in CR. For the sake of completeness for the reader, Table 2 summarizes the papers cited in this subsection.

## 4. Morphological Responses

Because of physical characteristics such as coat color, breeds respond differently to heat stress [91]. For instance, white coats are more tolerant to heat stress compared to animals with dark coats [91]. This is consistent with [92], in which the authors observed that animals with light pigmentation have substantially greater reflectance values for wavelengths between 300 and 850 nm than those with dark coats. A similar finding was reported in study [93], in which the authors observed that Boran cow body temperature gradients were lower than those of Nguni cows. This was most likely owing to increased skin temperature due to high external conditions resulting from vasocontrolled thermoregulation introducing more blood into the skin tissue [94]. This result was in agreement with work [95], the authors of which noted that lactating black cows have a body temperature increase of approximately 4.8 °C when exposed to direct sunshine, whereas it is about 0.7 °C for white cows. This results from the greater absorption of solar radiation by black coats (89%) as opposed to white coats (66%). Analogously, in [41], it was noticed that red and white cattle showed lower temperatures in white areas during the hot season compared to black and white animals. These findings imply that the red coat genotype absorbs less solar energy and then less heat. Deeply pigmented skin shields the underlying tissues from short-wave UV direct radiation by preventing its penetration [54].

Furthermore, coat length, thickness, and hair density influence the adaptation of animals in tropical climates, where short hair, thin skin, and fewer hair follicles per unit area are directly related to increased adaptability to hot temperatures [54]. Ref. [96] explored the association between Holstein cows’ survival rates and hair coat color in hot areas. The authors observed that cows living in the tropics must have a stronger aptitude for evaporative cooling, which is most likely associated to darker pigmentation, hair coat color and physical characteristics. Table 3 summarizes the papers cited in this subsection.

## 5. Behavioral Responses

The behavioral responses exhibited by dairy cattle in response to heat stress have been largely investigated and well established. During periods of elevated ambient temperatures, cows modify their actions to reduce the thermal load. These behavioral changes can manifest in various ways, including changes in dry matter and water intake, as well as changes in the duration and frequency of standing and lying events. Table 4 summarizes the papers that are cited in this section.

### 5.1. Standing and Lying Behavior

Standing and lying events are adaptation mechanisms that can aid both effective dissipation of body heat and heat load reduction from ground surfaces [97]. High temperatures may lead to prolonged standing behavior inducing a reduction in milk production [10,66]. By standing more frequently, cows may improve their respiratory efficiency and properly use their body surface area for a more efficient management of sensible and insensible heat loss, reducing the amount of heat, which they are able to transfer from a heated surface when they are lying [66]. Ref. [98] highlighted that standing during the first 21 days of a lactation can influence the subsequent lying behavior throughout the rest of the lactation period. Regardless of the heat stress level, ref. [38] claimed that difference in core body temperature is minimal, i.e., 0.07 °C, between standing and lying cows. According to the same study, cows were more likely to be lying during the warmest hours of the day, when their body temperature averaged the highest, than during the colder overnight hours. This result agrees with study [99], the authors of which investigated the lying behavior and activity in healthy grazing cows throughout the period from late gestation to early lactation. Cows exposed to cold and rainy weather spent less time lying down. Similarly, the authors of ref. [69] conducted an experimental campaign monitoring cow activity in order to discover behavioral changes induced by heat stress. The work concluded that, when heat stress is present, the amount of daily time spent resting decreases. Ref. [100] revealed a noticeable reduction in the daily lying time equal to 3 h when the THI increased from 56.2 to 73.8.

Consequently, cows dedicated less than 8 h per day to rest. With specific regard to air temperature and THI, ref. [101] observed that cows spend less time lying and more time standing when air temperature and THI increase. Lying time reduces by about 40 min when mean temperature is above 28 °C and by 48 min when the THI is higher than 72. These findings highlight the importance of monitoring cow behaviors and time spent lying and standing, these being measures useful indicators of cow adaptation to extreme environmental conditions and of their overall well-being.

### 5.2. Drinking Behavior

The drinking behavior of dairy cows is closely linked to their ability to cope with high temperatures, as water consumption increases with increasing temperature due to the need for evaporative cooling [102,103,104]. Ref. [104] observed that moving from a THI equal to 57 to a THI equal to 72, lactating cows drank 21% more. The correlation between drinking behavior and THI is further supported by study [105] where free-range dairy cows had the most drinking episodes when the THI was between 78 and 82. Drinking patterns were assessed over a span of three consecutive days. Each hour of the day underwent systematic categorization based on the THI: “normal” (THI < 70), “alert” (70 < THI < 78), “danger” (79 < THI < 82), “emergency” (THI > 82). Incorporating a standardized stress index, which quantifies various environmental indicators, can provide an easily interpretable metric for assessing stress levels. This index can complement decision support system by offering a concise measure of environmental data to monitor stress indicators. Moreover, ref. [106] observed that drinking frequency increased consistently, i.e., from 0.12 to 0.23 l/THI, throughout a range of THI increasing from 51.4 to 79.7. Additionally, changes in drinking patterns were observed between different troughs based on the dominance hierarchy of the cows, with a seasonal correlation of drinking time with THI [107]. Likewise, study [108] evaluated the ways in which animal behaviors alter in a subtropical location throughout the hot season, with or without unrestricted access to shaded area. Cows having access to shaded area had lower water ingestion episodes than cows without access to shade. These results were confirmed in [109], which examined the effects of heat stress on the behavior at the drinker in 69 indoor-housed lactating Holstein dairy cows. The reviewed literature agrees on the fact that cows in heat stress consume a greater amount of water, spend more time at the drinker, approach the drinker more frequently, and engage in more competitive interactions at the drinker.

### 5.3. Feeding Behavior

It is well known that heat stress leads dairy cows to reduce their daily feed intake, resulting in a decrease in milk yield and an increase in health issues. An indicator of heat stress in dairy cows is the decrease in appetite. Reduced feed intake is attributed to the discomfort caused by high temperature, as well as the energy required for sweat production, which is needed to regulate the cow’s body temperature [38,110]. In animals experiencing heat stress, a physiological adaptation to cope with internal metabolic heat production is the reduction in dry matter intake (DMI). In [54], significant differences were noticed in the feed sorting behavior of cows receiving evaporative cooling and those suffering from heat stress. Specifically cows under acute or chronic heat stress and without cooling opted for long ration particles, whereas cows receiving evaporative cooling did not sort for any proportion. Changes in behavior according to data gathered from the rumen environment may explain the consequences of heat stress [111]. Moreover, in the study conducted in ref. [62], cattle undergoing heat stress experienced a reduction in feed intake; this was observed in animals raised under thermoneutral conditions of 15% in the case of high-energy diet and 20% in the case of low-energy diet. Cows experiencing heat stress eat and ruminate less, which results in fewer buffering chemicals entering the rumen, where ruminating is the primary activator of saliva production [70]. According to [112], the mean daily DMI decreased less in heat-tolerant cows than in heat-susceptible cows. By decreasing the fermentation heat produced by the rumen, a reduction in daily DMI can be observed, which is a behavioral adaptation that decreases the generation of metabolic heat [113]. The additional metabolic heat production generated by dairy production may account for the more rapid adaptation observed in lactating cows [70,101]. Furthermore, a lack of DMI in transition dairy cows having higher energy need could result in an adverse energy balance [114]. Prior research has shown that heat stress can reduce de novo fatty acid synthesis in the mammary gland [115], which might cause acetate deficiency caused by a lower feed intake [116].

**Table 4 animals-13-03451-t004:** References related to behavioral responses in dairy cattle during heat stress.

Behavioral Responses	References	Section
Standing and lying behavior	(Shilja et al., 2016) [97]	Section 5.1
	(Polsky and von Keyserlingk, 2017) [10]	
	(Anderson et al., 2013) [66]	
	(Lovarelli et al., 2020) [98]	
	(Allen et al., 2015) [38]	
	(Hendriks et al., 2019) [99]	
	(Ramón-moragues et al., 2021) [69]	
	(Cook et al., 2007) [100]	
	(Hut et al., 2022) [101]	
Drinking behavior	(Hanušovský et al., 2017) [102]	Section 5.2
	(Sullivan and Mader, 2018) [103]	
	(Collier et al., 2019) [104]	
	(Pereyra et al., 2010) [105]	
	(Ammer et al., 2018) [106]	
	(Tsai et al., 2020) [107]	
	(Vizzotto et al., 2015) [108]	
	(McDonald et al., 2020) [109]	
Feeding behavior	(Allen et al., 2015) [38]	Section 5.3
	(Dourmad et al., 2022) [110]	
	(Sejian et al., 2018) [54]	
	(Miller-Cushon et al., 2019) [111]	
	(Meneses et al., 2021) [62]	
	(Sammad, Wang, et al., 2020) [70]	
	(Garner et al., 2016) [112]	
	(Bernabucci et al., 2009) [113]	
	(Hut et al., 2022) [101]	
	(Liu et al., 2017) [115]	
	(Urrutia et al., 2019) [116]	

## 6. Metabolic Responses

Animals have developed various strategies to cope with heat stress, and one of the most effective is to decrease the metabolic energy output through metabolic adaptation [117]. However, if heat stress persists for an extended period, it can lead to lethal heat stress. As previously mentioned, this condition can compromise the ability of the animals to thermoregulate, leading to hyperthermia, which can cause acute metabolic and systemic diseases [59]. The presence of hyperthermia and dehydration has been correlated with heightened neuromuscular fatigue and compromised movement coordination in animals. Hence, in regions characterized by hot climates, the potential for injury escalation is amplified [118]. Ref. [59] reported that lethal heat stress can result in a range of disorders, including endotoxemia, electrolyte imbalances, abnormalities of the acid-base system, and physiological disturbances including respiratory, cardiac, and renal failure. In vitro experimental models have demonstrated that a reducing tissue factor essentially limits inflammation-induced thrombin generation, thereby preventing endotoxemia [119]. However, the coagulation cascade can become widely activated by the tissue factor, generating thrombin, stimulating platelets, and producing platelet-–fibrin clots. Local perfusion consequences can lead to tissue hypoxia and organ failure caused by these microthrombi [120].

The conversion of feed into energy, for various metabolic activities such as milk production, maintenance, and development, is referred to as energetic metabolism in dairy cattle. Catabolism and anabolism are the two major processes involved in energetic metabolism. Under conditions of stress caused by high temperatures, lactating dairy cows experience a significant decrease in their blood glucose levels if compared to those maintained in thermal neutrality. This decrease primarily arises from a concomitant lower DMI, leading to a reduction in the overall glucose synthesis within the animal body [121,122]. Lactating cows that are subjected to heat stress exhibit high levels of plasma insulin, despite experiencing a decline in their blood glucose levels. This suggests that factors beyond blood glucose levels can impact the secretion of insulin in response to stressors [122,123]. Furthermore, several researchers have concluded that cows subject to heat stress show increased plasma urea nitrogen levels, indicating greater ammonia absorption in the gastrointestinal tract and enhanced hepatic elimination and amino acid deamination [124]. The results in [125] imply that the reduction in plasma amino acids measured during heat stress may be attributable to heightened amino acid consumption as gluconeogenic precursors. The occurrence of heat stress in lactating dairy cows has been shown to increase the quantity of hepatic glucose production [48]. With specific regard to the metabolism of proteins, ref. [126] revealed that heat stress led to the mobilization of skeletal muscle proteins, confirmed by an increase in blood creatine. These findings elucidate that short-term heat stress can enhance both adipose tissue lipolysis and skeletal muscle proteolysis as an energy source, whereas long-term heat stress inhibits adipose tissue lipolysis while intensifying skeletal muscle proteolysis. Table 5 summarizes the papers useful in this subsection.

## 7. Milk Responses

As previously mentioned, the ability of cows to dissipate heat is reduced by elevated temperatures and humidity, resulting in physiological and metabolic alterations that ultimately decrease milk yield and modify milk composition. Table 6 summarizes the papers cited in the following subsections.

### 7.1. Milk Yield

Several studies have shown the negative correlation between milk production and THI [15,34,127,128], particularly when the daily average THI overcomes the value of 68 [17]. The findings in [129] further supported this relationship, with an increase in THI leading to a decrease in daily milk production, especially in high-productivity cows. Additionally, the level of activity and milk yield decrease in response to heat load, particularly three [130,131] or five days [132] after the exposure. Ref. [133] found that the highest average milk yield was obtained in October while the lowest yield was recorded in July, thereby confirming the inverse correlation between milk yield and heat stress effects modelled with the THI. Similarly, ref. [134] observed that dairy cattle exposed to high temperatures and reducing the feed intake had a considerable drop, of about 20%, in the milk production compared to the part of the dairy cattle reared in optimal climatic conditions. In [135], it was observed that cattle that started the milking period during the hotter season of the year exhibited the lowest milk production, whereas cows that started the milking period in intermediate seasons demonstrated the highest milk yield. A recent investigation carried out in [136] revealed that during the final stage of pregnancy, an increase in the maternal body temperature could lead to a reduction in the amount of milk yield up to the three following lactations. The stage of lactation is a crucial factor in determining the impact of heat stress experienced by animals, since mid-lactation cows exhibit greater susceptibility compared to cows in early and late lactation stages. Additionally, during the early lactation stage, milk production in cows is largely sustained by mobilization of tissue stores and in a lower extent by feed intake. Conversely, in the mid-lactation stage, milk yield primarily relies on feed intake [137]. In this regard, past experimental tests have investigated the impact of reducing consumption of DMI on milk yield in animals affected by heat stress. It can be speculated that the direct influence of heat stress on milk yield indicates that a percentage ranging from 35% [138] to 50% [122] of the overall productivity drop can be attributed to a reduction in DMI [139]. Due to the higher efficiency of metabolic tissue store utilization in comparison with the metabolic utilization of feed, it is expected that early lactating cows would generate a lesser amount of metabolic heat per unit of produced milk with respect to mid-lactating cows [137]. This agrees with [139], the authors of which investigated cows in an intermediate phase of lactation, and reported an increased vulnerability to the adverse conditions associated to high THI recording modification of milk composition and a decline in milk production. Likewise, heat stress during the dry period undermines the development of the mammary gland prior to giving birth, leading to a decrease in milk production during the following lactation [140]. The thermal stress not only impacts milk quantity, but also adversely affects milk quality, leading to a decreased levels of proteins, fats and lactose [52].

### 7.2. Milk Quality

Several investigations [49,141,142] have confirmed the strong correlation between THI and percentage of both milk fat and milk protein. Ref. [144] noted a reduction in the concentration of fat and protein in milk during the summer months, whereas an increase in concentration was observed during the winter season, but on the other hand, it was found that lactose remained constant across the seasons. Contrary to this finding, in work [145], it was observed that the lactose level in grazing cattle decreases during the summer season compared to the winter season. Lactose is linked to the production of α-La, a coenzyme necessary for its formation. This specific whey protein is synthesized in the mammary gland, where α-La interacts with β-1,4-galactosyltransferase in the Golgi apparatus of the mammary gland epithelial cells to facilitate the formation of the lactose–synthase enzyme. As stated in [146], α-La alters the substrate specificity of β-1,4-galactosyltransferase to enable the creation of lactose from glucose and UDP-galactose.

With specific regard to milk fat, lipid represents a fundamental constituent, with a substantial portion consisting of polar lipids that act as emulsifiers and serve as the primary structural elements of the fat globule membrane. They are essential for the milk emulsion system to remain stable [147]. In [115], the content of polar lipids underwent a reduction of 43% following the exposure to heat stress for a duration of two days, subsequently decreasing further to 52% after four days from the day of the exposure. Following a five-day period of recovery, the polar lipid content rebounded to its baseline level. Furthermore, ref. [148] observed that cows who were exposed to high temperatures for one or more consecutive days produced less milk fat compared to animals that were not under heat stress conditions. With regard to the content of proteins in milk, the work revealed that the cows need to be stressed for at least three consecutive days to observe a significant reduction in protein content. There are various possible explanations for the decrease in milk protein, some of which may be inherit to the mammary gland. Ref. [134] revealed that the decrease in milk protein observed in heat stressed cows occurs due to a specific downregulation of protein synthesis in the mammary gland and not a byproduct of a general decrease in milk production. The authors determined that heat stress exclusively influenced the individual casein mass fraction proportions, leading to an increase in α_S1_-casein and a decrease in α_S2_-casein. Following [125], the authors of which investigated the impact of high temperatures on the generation of milk protein, cattle exposed to heat stress displayed a decrease in the total amount of amino acids in their plasma, which was lowered by 17.1%. Both essential and non-essential amino acids were impacted by this reduction.

## 8. Immune Status Responses

The immune response of dairy cattle is a complex process involving various cellular and molecular mechanisms that are affected by changes in environmental conditions. For instance, heat stress has been proven to decrease the activity and quantity of specific immune cells, such as neutrophils and lymphocytes. According to [148], heat stress is linked to a decrease in in vitro neutrophil phagocytosis and lymphocyte proliferation in crossbred cattle. These cells perform critical functions in battling infections and illness, and their decreased activity can make dairy cattle more susceptible to infectious pathogens. Likewise, the presence of heat stress causes a significant decrease in the percentage of CD21+MHCII+ B cell populations in both Holstein and Jersey steers. Furthermore, high temperature decreases the level of myeloperoxidase in polymorphonuclear cells in Holstein steers, whereas heat stress decreases the WC1 + γδ T cell populations in Jersey steers. Sahiwai, in comparison to Gir and Tharparkar breeds, exhibit elevated heat stress indicators, including reduced phagocytic activity, increased somatic cell count, heightened neutrophil population, elevated milk cortisol levels, and the activation of heat shock and cell adhesion markers [149]. In the mammary gland, neutrophils perform the critical role of phagocytosis and intracellular killing. This involves the process of chemoattraction, where neutrophils are drawn towards bacteria, followed by their engulfment using two distinct mechanisms, namely the respiratory burst and digestion using lysosomal enzymes [150]. In addition to the influence on immune cell function, heat stress may affect the synthesis of immune system elements such as cytokines and antibodies. Temperature increases has been demonstrated to reduce the production of cytokines, which are critical in immune response coordination. In [151], it is shown that heat stressed cows have reduced blood plasma concentrations of inflammatory cytokines and immunoglobulins (i.e., IgA, IgM, and IgG). As a result, the capacity of the immune system to mount a successful defense against pathogens may be compromised. According to these results, providing dairy cows with cooling measures during the initial period of the lactation has a positive impact on their production performance, metabolic indicators, immune response, and antioxidant capacity. Exposure to a heat stress environment alters physiological traits, which might also interact with bacterial metabolism and modify the systemic levels of cytokines, impacting the function of the brain–gut axis in dairy cows [152]. In comparison to cows maintained in thermoneutral conditions, ref. [153] reports that heat stressed cows express more IL-10. Similarly, ref. [154] notes that heat stress considerably affects the levels of IL-6 in the blood of cows during the interval preceding calving. These levels are much higher in heat-stressed animals compared to those in cows raised in non-stressing climate conditions. Furthermore, exposure to high temperatures can result in oxidative stress, a state in which the capability of the body to neutralize reactive oxygen species is out of balance with the amount of these species produced [59]. Oxidative stress may harm cells and tissues, as well as affect immune function by lowering immune cells capacity to operate in a proper way. In presence of a high THI level, mitochondria produce more superoxide anion. Additionally, the stress leads to an increase in the production of transition metal ions, which boosts the creation of superoxide anion [59]. Oxidative stress in dairy cows is caused by excessive reactive oxygen species (ROS) production and consequent mitochondrial dysfunction resulting from high temperatures. This oxidative stress, promoting both insulin resistance and apoptosis, which are negatively associated with milk protein synthesis, leads to a reduction in total milk protein [155]. Heat stress increases the risk of various diseases in dairy cattle, such as mastitis and lameness. High temperatures can weaken the immune system and make it more vulnerable to infections. Several studies have also evaluated the consequences of seasonal variations on the immunity of dairy cows. Based on a 12-month study conducted across 13 different herds, ref. [156] finds that the prevalence of Streptococcus dysgalactiae infections is lower at the beginning of the housing period compared to the pasture season [156]. Further, the impact of THI on female reproduction and health characteristics such as clinical mastitis, retained placenta, and puerperal disorders, is evaluated in 22,212 cows from Day 0 to Day 10 postpartum in [157], establishing that that the incidences rise along with a rising average THI. The immunological status responses of dairy cattle to heat stress involves several complicated pathways that are currently unknown. Nonetheless, it is known that heat stress can have a deleterious impact on dairy cattle’s immune systems, making them more prone to infections and illness. Table 7 summarizes the papers cited in this section.

## 9. Conclusions and Future Research Directions

This literature review highlights the critical need of assessing the physiological responses induced by heat stress in dairy cattle given its significant impact on both production sustainability and animal welfare. With growing awareness of the connectivity between animal health and resource efficiency, understanding the effects of heat stress on milk quantity and quality is critical in encouraging sustainable agriculture practices. The availability of large datasets in a quickly expanding technological landscape has opened access to advanced numerical techniques, such as machine learning and big data approaches, as well as in the livestock sector. The large amount of veterinary data available in the literature represents a huge potential that should not be wasted. The common patterns and information gaps are revealed by a comprehensive investigation of numerous features, including physiological, morphological, behavioral, metabolic, productive, and immunological responses. This approach involves searches across scientific databases, careful selection of pertinent papers, and comprehensive evaluation of the quality and reliability of the findings. This review provides a complete perspective by concentrating on statistically determined effects of heat stress on lactation cows, evaluating current research gaps and collecting essential veterinary knowledge from the literature. The review consolidates and synthesizes existing knowledge on the physiological indicators of heat stress in dairy cattle to properly address the development of numerical models and decision support systems. Ultimately, this comprehensive synthesis not only emphasizes the critical role of physiological insights in advanced numerical approaches, but also highlights the importance of multidisciplinary collaboration between veterinary research and emerging technologies. Therefore, the present work must be seen as one of the pieces of the puzzle and in a broader interdisciplinary context and therefore needs to be related to the specific literature of the other research sectors. Thus, in this case, the information organized in the present review is preparatory to the definition of the most suitable numerical tools. Finally, numerical techniques may be a valuable technique for predicting heat stress cases, but it takes a large amount of data to produce accurate predictions. ML models, as well as the use of sensors, can be an effective combination to obtain a large volume of useful information on heat stress conditions, which may aid in overcoming the actual issues related to the use of ML models. Furthermore, constant monitoring of several indicators connected with animal well-being, activities and milk production may be valuable for improving the performance of these instruments. As climate change continues to present challenges, this review may be used to promote resilience and ensure the well-being of dairy cattle and the dairy industry.

## Figures and Tables

**Figure 1 animals-13-03451-f001:**
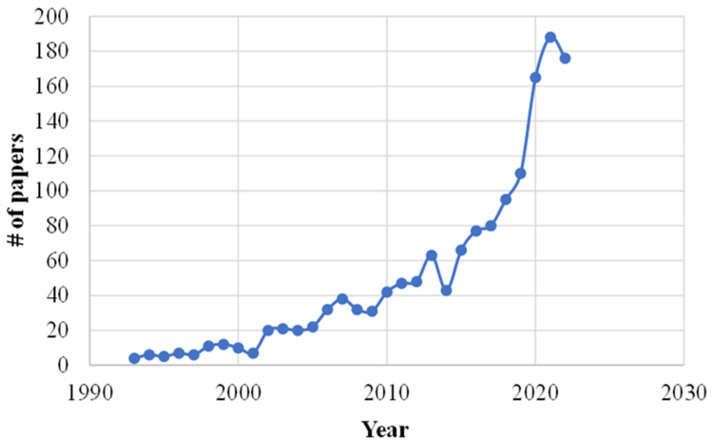
Trend of the bibliography sources investigating heat stress on dairy cows.

**Figure 2 animals-13-03451-f002:**
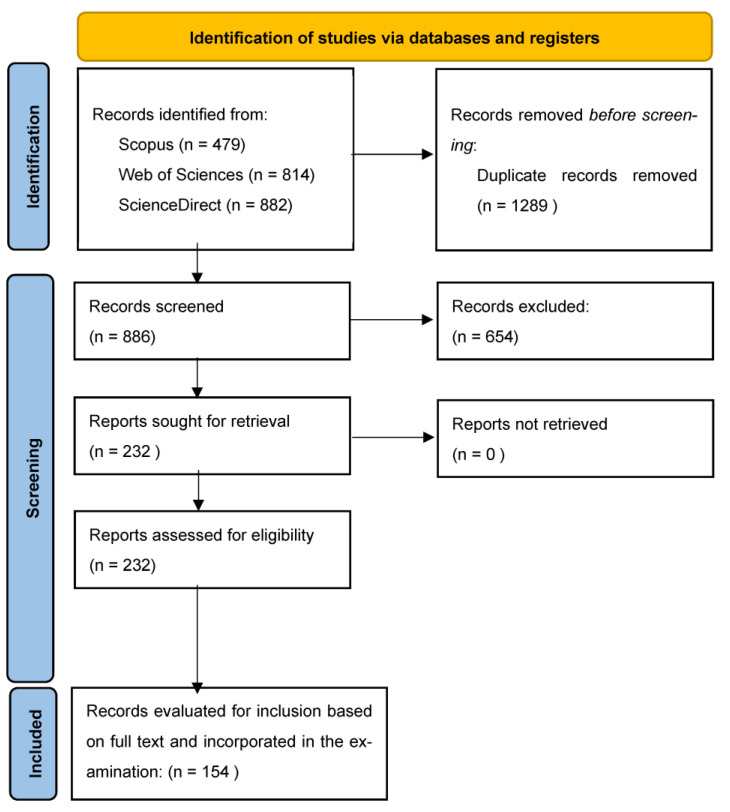
Flow diagram of the bibliographic source selection process. From Page et al. [44].

**Figure 3 animals-13-03451-f003:**
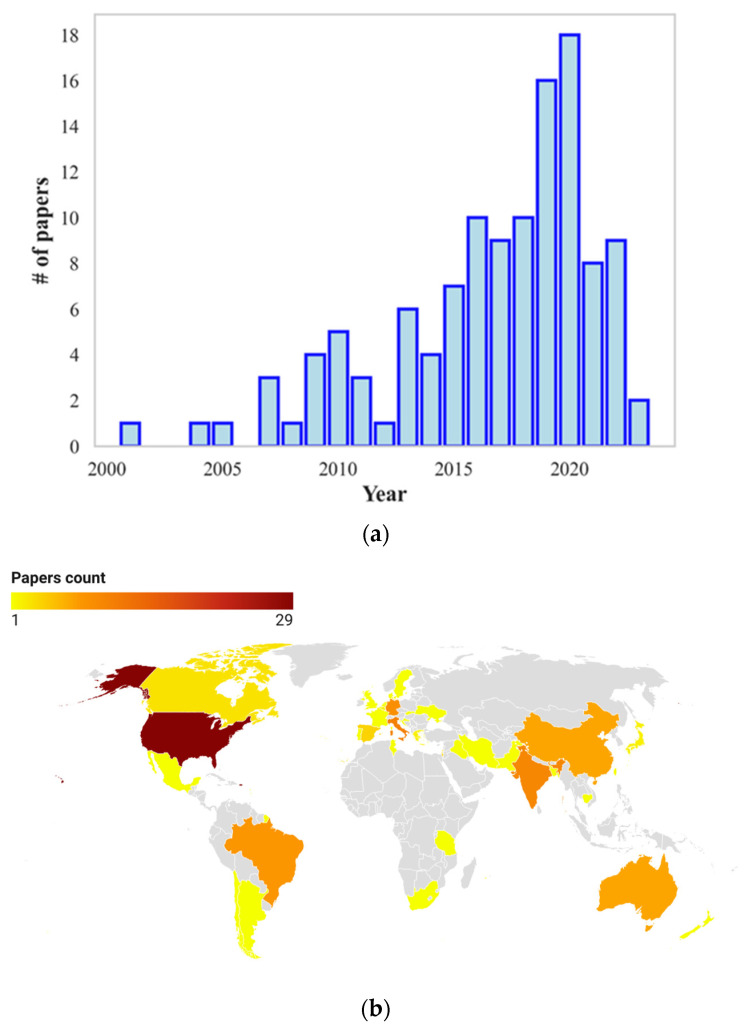
Papers selected for the present review. (**a**) Timeline of the number of papers; (**b**) Geographical distribution.

**Figure 4 animals-13-03451-f004:**
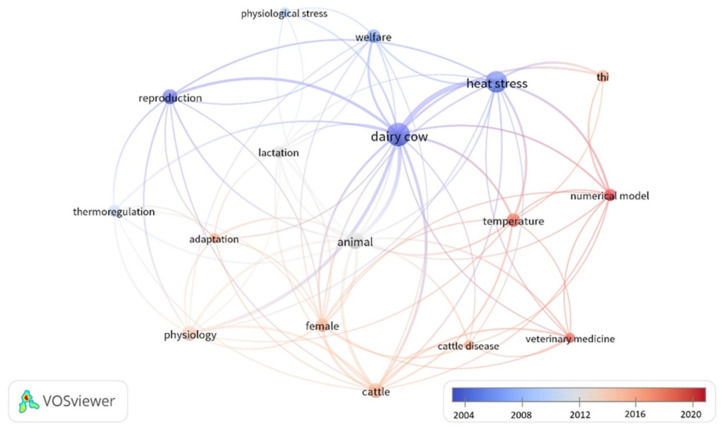
Temporal network map of the keywords most frequently used by the 19 review papers. The dimension of the node increases with the number of occurrences while the different colors indicate the years with most of the occurrences.

**Table 1 animals-13-03451-t001:** Different measure classes used for assessing animal welfare.

Measure Class	Description	Advantages	Limitations
Animal-based measures (AMB)	Methods for assessing animal welfare that inspect how animals react physiologically and behaviorally to their surroundings and encounters (e.g., body condition score, milk production and composition, reproductive performance, time spent lying, etc.).	Provide precise and immediate quantification.	Require in-depth knowledge and understanding of the animal.
Resource-based measures (RBM)	Animal welfare indicators that evaluate the availability and quality of resources provided to animals (e.g., housing conditions, lighting, ventilation efficiency, access to shade and pasture, etc.).	Provide objective and quantitative data.	Focusing on the physical structures does not allow evaluation of the welfare of the individual animal.
Management-based measures (MBM)	Animal welfare measures that evaluate how animals are managed or treated, including the management practices employed in rearing animals and the physical environment in which they are housed (e.g., training and education, implementation of animal welfare policies, record-keeping and data analysis).	Provide practical solutions to welfare issues.	May not reflect animal welfare. Can be affected by human biases.

**Table 2 animals-13-03451-t002:** References related to physiological responses in dairy cattle during heat stress.

Physiological Responses	References	Section
Body temperature and respiration rate	(Neves et al., 2022) [60]	Section 3.1
	(Rathwa et al., 2017) [61]	
	(Idris et al., 2021) [8]	
	(Meneses et al., 2021) [62]	
	(Jeelani et al., 2019) [63]	
	(Scharf et al., 2010) [64]	
	(Sejian et al., 2018) [54]	
	(Burfeind et al., 2012) [18]	
	(van Os, 2019) [65]	
	(Anderson et al., 2013) [66]	
	(Ji et al., 2020) [58]	
	(Wang et al., 2020) [67]	
	(Grotjan et al., 2020) [68]	
	(Ramón-moragues et al., 2021) [69]	
	(Polsky and von Keyserlingk, 2017) [10]	
	(Pinto et al., 2020) [7]	
	(Sammad et al., 2020) [88]	
Heart rate and rumination time	(Kovács et al., 2018) [71]	Section 3.2
	(Bun et al., 2018) [72]	
	(Meneses et al., 2021) [62]	
	(Cardoso et al., 2015) [73]	
	(Lakhani et al., 2018) [74]	
	(Grinter et al., 2023) [75]	
	(Toledo et al., 2022) [76]	
	(Stone et al., 2017) [77]	
	(Moretti et al., 2017) [78]	
	(Müschner-Siemens et al., 2020) [13]	
	(Soriani et al., 2013) [79]	
	(Ramón-moragues et al., 2021) [69]	
	(Maia et al., 2020) [12]	
Reproduction and fertility	(Wolfenson and Roth, 2019) [80]	Section 3.3
	(Nanas et al., 2021) [81]	
	(Roy and Prakash, 2007) [82]	
	(Lucy, 2001) [83]	
	(Roth, 2017) [51]	
	(Stamperna et al., 2020) [84]	
	(Piccioni et al., 2005) [85]	
	(Gendelman et al., 2010) [86]	
	(Hansen, 2007) [87]	
	(Sammad et al., 2020) [88]	
	(Roth, 2020) [89]	
	(Nabenishi et al., 2011) [90]	
	(Schüller et al., 2014) [30]	

**Table 3 animals-13-03451-t003:** References related to morphological responses in dairy cattle during heat stress.

Morphological Responses	References	Section
Coat color and heat stress	(McManus et al., 2009) [91]	Section 4
	(da Silva et al., 1962) [92]	
	(Katiyatiya et al., 2017) [93]	
	(Soerensen and Pedersen, 2015) [94]	
	(P. E Hillman et al., 2013) [95]	
	(Isola et al., 2020) [41]	
	(Sejian et al., 2018) [54]	
	(Lee et al., 2016) [96]	

**Table 5 animals-13-03451-t005:** References related to metabolic responses in dairy cattle during heat stress.

Metabolic Responses	References	Section
Energetic metabolism	(Pragna et al., 2018) [117]	Section 6
	(Burhans et al., 2022) [59]	
	(Temple et al., 2020) [118]	
	(Van Der Poll et al., 2008) [119]	
	(Gyawali et al., 2019) [120]	
	(Lamp et al., 2015) [121]	
	(Wheelock et al., 2010) [122]	
	(Koch et al., 2016) [123]	
	(Ríus A.G., 2019) [124]	
	(Gao et al., 2017) [125]	
	(Baumgard et al., 2013) [48]	
	(Hou et al., 2021) [126]	

**Table 6 animals-13-03451-t006:** References related to milk production responses in dairy cattle during heat stress.

Milk Responses	References	Section
Milk yield	(M’Hamdi et al., 2021) [127]	Section 7.1
	(Kino et al., 2019) [15]	
	(Ekine-Dzivenu et al., 2020) [34]	
	(Gauly et al., 2013) [128]	
	(Tao et al., 2020) [17]	
	(Summer et al., 2019) [129]	
	(Heinicke et al., 2019) [130]	
	(Wildridge et al., 2018) [131]	
	(Manica et al., 2022) [132]	
	(Reyad et al., 2016) [133]	
	(Cowley et al., 2015) [134]	
	(Mellado et al., 2011) [135]	
	(Laporta et al., 2020) [136]	
	(Bernabucci et al., 2010) [137]	
	(Rhoads et al., 2009) [138]	
	(Wheelock et al., 2010) [122]	
	(Moore et al., 2023) [139]	
	(Tao et al., 2011) [140]	
	(Prathap et al., 2016) [52]	
Milk quality	(Das et al., 2016) [49]	Section 7.2
	(Mylostyvyi et al., 2019) [141]	
	(Bertocchi et al., 2014) [142]	
	(Hill and Wall, 2014) [143]	
	(Bernabucci et al., 2015) [144]	
	(Florio et al., 2022) [145]	
	(Farrell et al., 2004) [146]	
	(Sánchez-Juanes et al., 2009) [147]	
	(Liu et al., 2017) [115]	
	(Ouellet et al., 2019) [148]	
	(Cowley et al., 2015) [134]	
	(Gao et al., 2017) [125]	

**Table 7 animals-13-03451-t007:** References related to immune status responses in dairy cattle during heat stress.

Immune Status Responses	References	Section
Immunity	(Tejaswi et al., 2020) [148]	Section 8
	(Alhussien et al., 2018) [149]	
	(Alhussien et al., 2016) [150]	
	(Safa et al., 2019) [151]	
	(Chen et al., 2018) [152]	
	(Thompson et al., 2014) [153]	
	(Ali Judi et al., 2022) [154]	
	(Burhans et al., 2022) [59]	
	(Guo et al., 2021) [155]	
	(Lundberg et al., 2016) [156]	
	(Gernand et al., 2019) [157]	

## Data Availability

No new data were created in this study. Data sharing is not applicable to this article.

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
