# Peer review of "(untitled)"

_animals, 2023, doi:10.3390/ani13223451_

Round 1

Reviewer 1 Report

Comments and Suggestions for Authors

General Comment

Before submission, authors need to check the Checklist:

- Review the structure and organization of the article (Introduction with clear objectives, Methodology that allows replication, Results with robust analysis, and Conclusions that meet the objectives).

- Verify if the data analysis methods are well explained in the methodology.

- Tables and figures should be self-explanatory and correctly mentioned in the text.

- Is the discussion current and sufficient?

- Make sure the conclusions highlight the research.

- Check the quality and relevance of the references and if the scope of the work is aligned with the scope of this Journal.

- Check that all references are in the text

Strengths:

The topic is relevant, exploring a bibliographic review to record emerging technologies that have emerged as a decision-making support tool.

There is an extensive bibliographical review. The methodology describes that 120 references were gathered, and the article contains 155 bibliographies on the subject. It's quite a lot, but I found it confusing.

The sections are well organized and follow a logical structure.

 Weaknesses:

Some paragraphs are very long and tiring. As they mix topics, they can be separated into different paragraphs.

Several bibliographies are cited superficially, without comparison with others, and without discussing their results to justify their presence. Therefore, I suggest a greater critical integration of the literature, discussing convergences and divergences between the studies reviewed.

Just as it included a study on the THI, as it is a review work, it is recommended to compare the main thermal comfort indices to establish which is the most suitable for evaluating thermal stress levels in dairy cattle farming. At least THI, Enthalpy and BGHI.

Have any quantitative analyses of previous results been carried out (meta-analysis)? It was a little superficial how the references were chosen, mainly because there were few from 2023.

There was a lack of conclusion on using an index, together with decision-making support systems, for the correct assessment of stress levels.

It remained to highlight the gaps that were not addressed in the study and research possibilities that will benefit from this review.

If one of the objectives of this review is to highlight the importance of new analysis, monitoring, algorithms and AI technologies, there was a topic missing from this content.

 On page 1 (Introduction), lines 42 and 43, of the six references, only one is current. We are already approaching 2024!

Suggestion: [1-4] which are from the years [2013, 2018, 2022, 2003] becomes [2018, a, 2022]

a - Cesca, R. S., Santos, R. C., Goes, R. H. D. T., Favarim, A. P. C., Oliveira, M. S. G. D., & Silva, N. C. D. (2021). Thermal comfort of beef cattle in the state of Mato Grosso do Sul, Brazil. Ciência e Agrotecnologia, 45. https://doi.org/10.1590/1413-7054202145008321

Suggestion: [5, 6] which are from the years [2014, 2002] becomes [2014, b]

b- Santos, R. C., Lopes, A. L., Sanches, A. C., Gomes, E. P., da Silva, E. A., & da Silva, J. L. (2023). Intelligent automated monitoring integrated with animal production facilities. Engenharia Agrícola, 43, e20220225. https://doi.org/10.1590/1809-4430-Eng.Agric.v43n2e20220225/2023

 The review on THI lines 334 to 340 is very superficial. If the reader is not familiar with the topic, they will not know what THI is or what the stress and comfort ranges are. 

Author Response

General Comment

R) We checked and verified the "General Comment" part.

Strengths

R) We thank the Reviewer for the appreciations.

Weaknesses

Reviewer 2 Report

Comments and Suggestions for Authors

General Comments

The paper titled "Review of the heat stress-induced responses in dairy cattle" aims to provide a comprehensive overview of the physiological responses of dairy cattle to heat stress, with a specific focus on making this knowledge accessible to researchers and technicians developing numerical models and decision support tools. The main contributions of this review lie in its attempt to consolidate and synthesize existing research on the topic. However, some concerns have been raised in the comments. The authors should clarify how it differs from existing review articles on the same topic and emphasize its novel contributions. Additionally, the authors are encouraged to address the limitations of their research and provide new insights or ideas rather than serving as a mere compilation of existing knowledge. It's important to highlight why their work is significant compared to existing publications in the field.

Round 2

Reviewer 2 Report

Comments and Suggestions for Authors

Dear Authors,

I have reviewed your manuscript and commend the revisions made according to the comments and suggestions provided during the first revision. I appreciate your thorough attention to detail in addressing the comments, which significantly improved the quality and clarity of the manuscript. Based on the revisions, I am pleased to accept the article for publication. Thank you for your efforts.